# THOMAS: Trajectory Heatmap Output with learned Multi-Agent Sampling

**Thomas Gilles**[1,2]**, Stefano Sabatini**[1]**, Dzmitry Tsishkou**[1]**, Bogdan Stanciulescu**[2]**, Fabien Moutarde**[2]

[1]IoV team, Paris Research Center, Huawei Technologies France [2]Center for robotics, MINES ParisTech

`{stefano.sabatini, dzmitry.tsishkou}@huawei.com`
`{thomas.gilles, bogdan.stanciulescu, fabien.moutarde}@mines-paristech.fr`

## Abstract

In this paper, we propose THOMAS, a joint multi-agent trajectory prediction framework allowing for an efficient and consistent prediction of multi-agent multi-modal trajectories. We present a unified model architecture for simultaneous agent future heatmap estimation, in which we leverage hierarchical and sparse image generation for fast and memory-efficient inference. We propose a learnable trajectory recombination model that takes as input a set of predicted trajectories for each agent and outputs its consistent reordered recombination. This recombination module is able to realign the initially independent modalities so that they do no collide and are coherent with each other. We report our results on the Interaction multi-agent prediction challenge and rank $1^{st}$ on the online test leaderboard.

## 1 Introduction

Motion forecasting is an essential step in the pipeline of an autonomous driving vehicle, transforming perception data into future prediction which are then leveraged to plan the future moves of the autonomous cars. The self-driving stacks needs to predict the future trajectories for all the neighbor agents, in a fast and coherent way.

The interactivity between agents plays an important role for accurate trajectory prediction. Agents need to be aware of their neighbors in order to adapt their speed, yield right of way and merge in neighbor lanes. To do so, different interaction mechanisms have been developed, such as social pooling (Alahi et al., 2016; Lee et al., 2017; Deo & Trivedi, 2018), graphs (Salzmann et al., 2020; Zeng et al., 2021) or attention (Mercat et al., 2020; Messaoud et al., 2020; Luo et al., 2020; Gao et al., 2020; Liang et al., 2020), benefiting from the progress of powerful transformer architectures (Li et al., 2020; Yuan et al., 2021; Girgis et al., 2021; Ngiam et al., 2021) . These mechanisms allow agents to look at and share features with neighbors and to take them into account in their own predictions.

Multi-modality is another important aspect of the possible future trajectories. A car can indeed chose to turn right or left, or decide to realise a certain maneuver in various ways. Uncertainty modeled as variance of Gaussians is insufficient to model these multiple cases, as it can only represent a continuous spread and cannot show multiple discrete possibilities. Therefore, current state-of-the-art produces not one but $K$ possible trajectories for each agent predicted, and most recent benchmarks (Caesar et al., 2020; Chang et al., 2019; Zhan et al., 2019; Ettinger et al., 2021) include multi-modality in their metrics, taking only the minimum error over a predicted set of $K$ trajectories.

However, up until very recently and the opening of multi-agent joint interaction challenges (Ettinger et al., 2021; Zhan et al., 2021), no motion forecasting prediction datasets were taking into account the coherence of modalities between different agents predicted at the same time. As a result, the most probable predicted modality of a given agent could crash with the most probable modality of another agent.

Our THOMAS model encodes the past trajectories of all the agents present in the scene, as well as the HD-Map lanelet graph, and then predicts for each agent a sparse heatmap representing the future probability distribution at a fixed timestep in the future, using hierarchical refinement for very efficient decoding. A deterministic sampling algorithm then iteratively selects the best K trajectory

endpoints according to the heatmap for each agent, in order to represent a wide and diverse spectrum of modalities. Given this wide spectrum of endpoints, a recombination module takes care of addressing consistency in the scene among agents.

Our contributions are summarized as follow:

- We propose a hierarchical heatmap decoder allowing for unconstrained heatmap generation with optimized computational costs, enabling efficient simultaneous multi-agent prediction.

- We design a novel recombination model able to recombine the sampled endpoints to obtain scene-consistent trajectories across the agents.

## 2 RELATED WORK

Learning-based models have quickly overtaken physics-based methods for trajectory prediction for several reasons. First, the sequential nature of trajectories is a logical application for recurrent architectures (Alahi et al., 2016; Altché & de La Fortelle, 2017; Lee et al., 2017; Mercat et al., 2020; Khandelwal et al., 2020). Then, benefiting from the latest progresses in computer vision, convolutional layers can easily be applied to bird-view rasters of the map context (Lee et al., 2017; Tang & Salakhutdinov, 2019; Cui et al., 2019; Hong et al., 2019; Salzmann et al., 2020; Chai et al., 2020; Gilles et al., 2021b). A more compact representation closer to the trajectory space can encode surrounding HD-Maps (usually formalized as connected lanelets), using Graph Neural Networks (Gao et al., 2020; Liang et al., 2020; Zeng et al., 2021; Gilles et al., 2021a). Finally, some point-based approaches (Ye et al., 2021) can be applied in a broader way to trajectory prediction, as both lanes and trajectories can be considered as ordered set of points.

Multi-modality in prediction can be obtained through a multiple prediction head in the model (Cui et al., 2019; Liang et al., 2020; Ngiam et al., 2021; Deo et al., 2021). However, some methods rather adopt a candidate-based approach where potential endpoints are obtained either from anchor trajectories through clustering (Chai et al., 2020; Phan-Minh et al., 2020) or from a model-based generator (Song et al., 2021). Other approaches use a broader set of candidates from the context graph (Zhang et al., 2020; Zhao et al., 2020; Zeng et al., 2021; Kim et al., 2021) or a dense grid around the target agent (Deo & Trivedi, 2020; Gu et al., 2021; Gilles et al., 2021b;a). Another family of approaches use variational inference to generate diverse predictions through latent variables (Lee et al., 2017; Rhinehart et al., 2018; Tang & Salakhutdinov, 2019; Casas et al., 2020) or GAN (Gupta et al., 2018; Rhinehart et al., 2018; Sadeghian et al., 2019) but the sampling of these trajectories is stochastic and does not provide any probability value for each sample.

While very little work has directly tackled multi-agent prediction and evaluation so far, multiple methods hint at the ability to predict multiple agents at the same time (Liang et al., 2020; Ivanovic et al., 2020; Zeng et al., 2021) even if they then focus on a more single-agent oriented framework. Other works (Alahi et al., 2016; Tang & Salakhutdinov, 2019; Rhinehart et al., 2019; Girgis et al., 2021) use autoregressive roll-outs to condition the future step of an agent on the previous steps of all the other agents. SceneTransformer (Ngiam et al., 2021) repeats each agent features across possible modalities, and performs self-attention operations inside each modality before using a loss computed jointly among agents to train a model and evaluate on the WOMD dataset (Ettinger et al., 2021) interaction track. ILVM (Casas et al., 2020) uses scene latent representations conditioned on all agents to generate scene-consistent samples, but its variational inference does not provide a confidence score for each modality, hence LookOut (Cui et al., 2021) proposes a scenario scoring function and a diverse sampler to improve sample efficiency. $AIR^2$ (Wu & Wu, 2021) extends Multipath (Chai et al., 2020) and produces a cross-distribution for two agents along all possible trajectory anchors, but it scales exponentially with the number of agents, making impractical for a real-time implementation that could encounter more than 10 agents at the same time.

The approach most related to this paper is GOHOME (Gilles et al., 2021a), which uses a similar graph encoder and then leverage lane rasters to generate a probability heatmap in a sparse manner. However, this lane-generated heatmap remains constrained to the drivable area and to fixed lane-widths. Another close approach is the one of DenseTNT (Gu et al., 2021), which also uses attention to a grid of points in order to obtain a dense prediction, but their grid also remains constrained to a

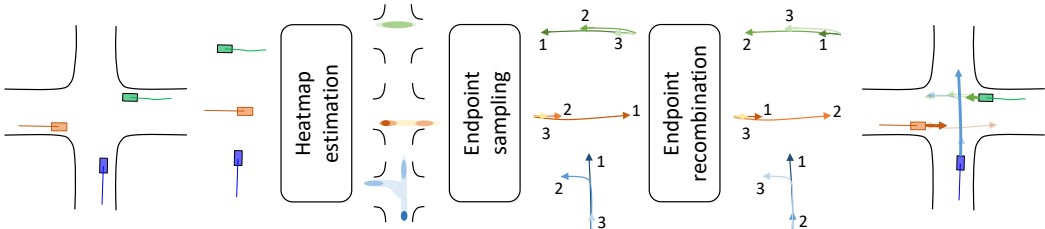

Figure 1: Illustration of the THOMAS multi-agent prediction pipeline

neighborhood of the drivable area. Finally, none of these previous methods considers the problems of scene consistency for multi-agent prediction.

## 3 METHOD

Our goal is to predict the future $T$ timesteps of $A$ agents using their past history made of $H$ timesteps and the HD-Map context. Similar to recent works (Zhao et al., 2020; Zeng et al., 2021; Gu et al., 2021), we will divide the problem into goal-based prediction followed by full trajectory reconstruction. Our prediction pipeline is displayed in Fig. 1. We first encode each agent trajectory and the HD-Map context graph into a common representation. We then decode a future probability heatmap for each agent in the scene, which we sample heuristically to maximize coverage. Finally, we recombine the sampled endpoints into scene-consistent modalities across agents and build the full trajectories for each agent.

Our pipeline shares the same graph encoder, sampling algorithm and full trajectory generation as GOHOME (Gilles et al., 2021a), but uses a novel efficient hierarchical heatmap process that enables to scale to simultaneous multi-agent prediction. Furthermore, we add a novel scene-consistency module that recombines the marginal outputs into a joint prediction.

### 3.1 MODEL BACKBONE

#### 3.1.1 GRAPH ENCODER

We use the same encoder as the GOHOME model (Gilles et al., 2021a). The agent trajectories are encoded though $TrajEncoder$ using a 1D CNN followed by a UGRU recurrent layer (Rozenberg et al., 2021), and the HD-Map is encoded as a lanelet graph using a GNN $GraphEncoder$ made of graph convolutions. We then run cross-attention $Lanes2Agents$ to add context information to the agent features, followed by self-attention $Agents2Agents$ to observe interaction between agents. The final result is an encoding $F_a$ for each agent, where history, context and interactions have been summarized. This encoding $F_a$ is used in the next decoder operations and is also stored to be potentially used in modality recombination described in Sec. 3.2. The resulting architecture of these encoding operations is illustrated in the first half of Fig. 2.

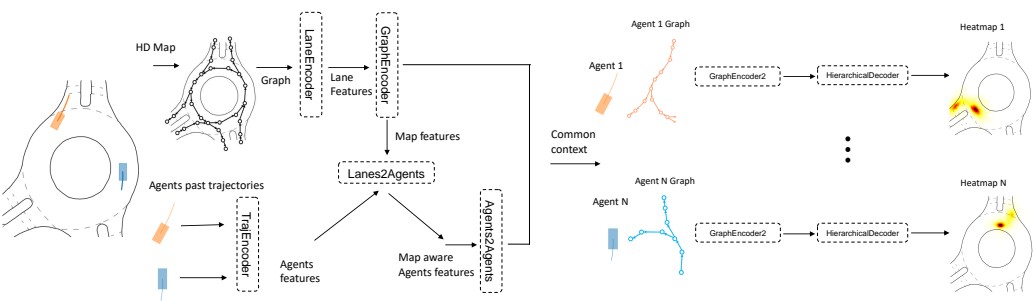

Figure 2: Model architecture for multi-agent prediction with shared backbone

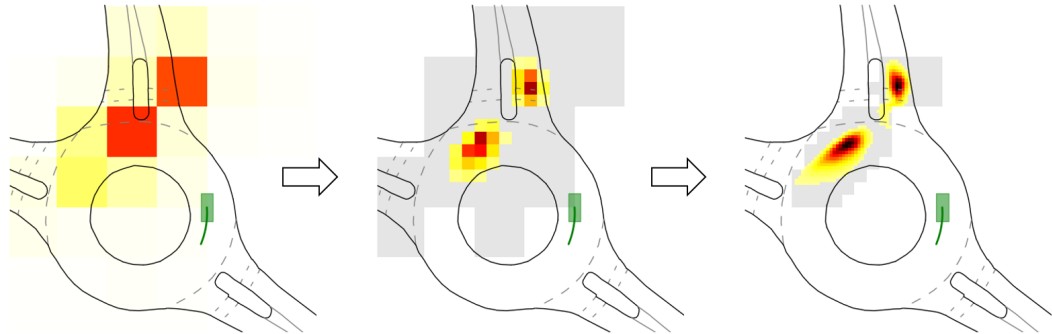

Figure 3: Hierarchical iterative refinement of the grid probabilities. First, the full grid is evaluated at a very low resolution, then the highest cells are up-sampled and evaluated at a higher resolution, until final resolution is reached. We highlight in grey the restricted area considered for refinement at each step.

### 3.1.2 HIERARCHICAL GRID DECODER

Our aim here is to decode each agent encoding into a heatmap representing its future probability distribution at prediction horizon $T$. Since we create this heatmap for each agent in the scene, the decoding process has to be fast so that it can be applied to a great number of agents in parallel. We use hierarchical predictions at various levels of resolutions so that the decoder has the possibility of predicting over the full surroundings of the agent but learns to refine with more precision only in places where the agent will end up with high probability. This hierarchical process is illustrated in Fig. 3.

Starting from an initial full dense grid probability at low resolution $R_0 \times R_0$ by pixels, we iteratively refine the resolution by a fixed factor $f$ until we reach the desired final resolution $R_{final} \times R_{final}$. At each iteration $i$, we select only the $N_i$ highest ranking grid points from the previous iteration, and upsample only these points to the $R_i \times R_i = {R_{i-1}}/f \times {R_{i-1}}/f$. At each step, the grid points features are computed by a 2-layer MLP applied on the point coordinates, these features are then concatenated to the agent encoding and passed through a linear layer, and finally enriched by a 2-layer cross-attention on the graph lane features, before applying a linear layer with sigmoid to get the probability.

For a given $W$ output range, this hierarchical process allows the model to only operate on ${W}/{R_0} \times {W}/{R_0} + \sum_i N_i \times f^2$ grid points instead of the ${W}/{R_{final}} \times {W}/{R_{final}}$ available. In practice, for a final output range of 192 meters with desired $R_{final} = 0.5m$ resolution, we start with an initial resolution of $R_0 = 8m$ and use two iterations of $(N_1, N_2) = (16, 64)$ points each and an upscaling factor $f = 4$. This way, we compute only 1856 grid points from the 147 456 available, with no performance loss.

The heatmap is trained on each resolution level using as pixel-wise focal loss as in Gilles et al. (2021b), detailed in Appendix A.3, with the ground truth being a Gaussian centered at the target agent future position.

### 3.1.3 FULL TRAJECTORY GENERATION

Having a heatmap as output grants us the good property of letting us decide to privilege coverage in order to give a wide spectrum of candidates to the scene recombination process. From each heatmap, we therefore decode $K$ end points using the same MissRate optimization algorithm as Gilles et al. (2021b). We then generate the full trajectories for each end point using also the same model, a fully-connected MLP. The MLP takes the car history and predicted endpoint as input and produces $T$ 2D-coordinates representing the full future trajectory. At training time, this model is trained using ground-truth endpoints.

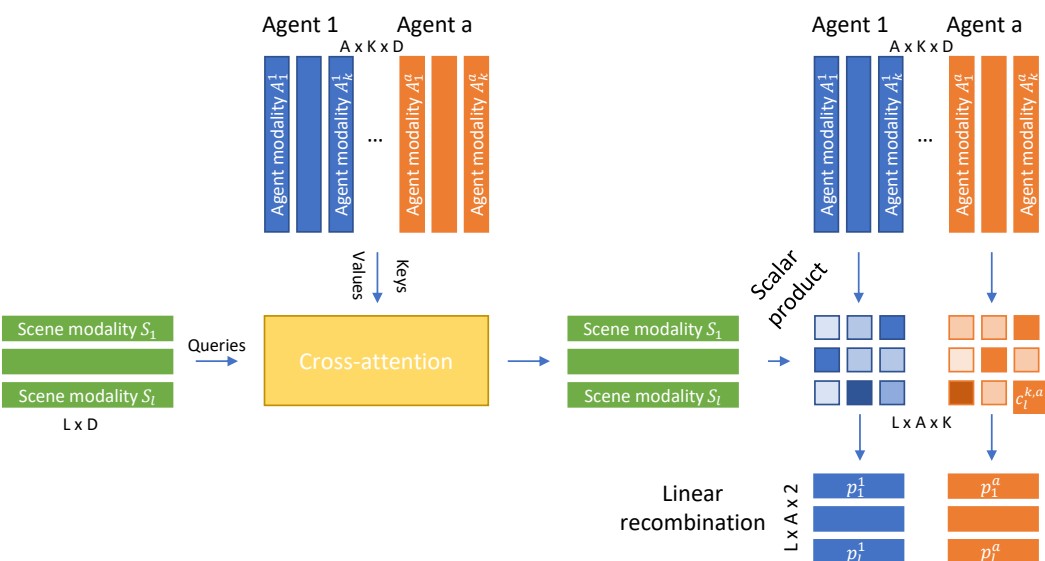

Figure 4: Illustration of THOMAS methods for generation of scene-consistent agent modalities

## 3.2 MODALITY RECOMBINATION FOR MULTI-AGENT CONSISTENT PREDICTION

The difficulty in multimodal multi-agent prediction resides in having coherent modalities between each agent. Since the modalities are considered scene-wise, each agent's most probable modality has to match with the most probable prediction of the other agents, and so on. Moreover, these modalities must not collide with each other, as they should represent realistic scenarios.

Initially, the prediction output of the model can be defined as marginal, as all $A$ agents have been predicted and sampled independently in order to get $K$ possible endpoint each. Our goal is to output a set of scene predictions $\mathcal{J} = (L, A)$ from the marginal prediction $\mathcal{M} = (A, K)$, where each scene modality $l$ belonging to $\mathcal{J}$ is an association of endpoints $p_l^a$ for each agent $a$. To achieve this, our main hypothesis lays in the fact that good trajectory proposals are already present in the marginal predictions, but they need to be coherently aligned among agents to achieve a set of overall consistent scene predictions. For a given agent $a$, the modality selected for the scene $l$ would be a combination of this agent available marginal modalities $p_k^a \in \mathcal{M}$. Each scene modality $l$ would select a different association between the agents.

We illustrate our scene modality generation process in Fig. 4. The model learns $L$ scene modality embeddings $S_l$ of $D$ features each. $K \times A$ agent modality vectors $A_k^a$ are also derived from each agent modality endpoint. These vectors are obtained through a 2-layer MLP applied on the agent modality coordinates $p_k^a$, to which the stored agent encoding $F_a$ (previously described in Sec. 3.1.1) is concatenated in order to help the model recognise modalities from the same agent. The scene modality vectors $S_l$ are 'specialized' by querying the available modality proposals $A_k^a$ of each agent through cross-attention layers. Subsequently, a matching score $c_l^{k,a}$ between each scene modality $S_l$ and each agent modality $A_k^a$ is computed. This matching score can be intuitively interpreted as the selection of the role (maneuver $k$) that the agent $a$ would play in the overall traffic scene $l$:

$$c_l^{k,a} = S_l . A_k^{a T}$$

Since argmax is non-differentiable, we employ a soft argmax as a weighted linear combination of the agent modalities $p_k^a$ using a softmax on the $c_l^{k,a}$ scores:

$$p_l^a = \sum_k softmax(c_l^{k,a}) p_k^a$$

The recombination module is trained with the scene-level minimum displacement error, where for each modality inside a single scene all predicted agent displacement losses are averaged before taking the minimum (winner-takes-all loss) over the $L$ scene modalities. A formal definiton of this loss, also used as a metric, is given in Eq. 2 in Sec. 4.2 (minSFDE$_k$).

## 4 EXPERIMENTS

### 4.1 DATASET

We use the Interaction v1.2 dataset that has recently opened a new multi-agent track in the context of its Interpret challenge. It contains 47 584 training cases, 11 794 validation cases and 2 644 testing cases, with each case containing between 1 and 40 agents to predict simultaneously.

### 4.2 METRICS

For a set of $k$ predictions $p_k^a$ for each agent $a$ in a scene, we report the minimum Final Displacement Error (minFDE$_k$) to the ground truth $\hat{p}^a$ and the MissRate (MR$_k$), whose definition of a miss is specified according to the evaluated dataset in Appendix A.1. In their usual marginal definition, these metrics are averaged over agents after the minimum operation, which means that the best modality of each agent is selected independently for each and then averaged:

$$minFDE_k = \frac{1}{A}\sum_a min_k\|p_k^a - \hat{p}^a\|_2 \, , \; MR_k = \frac{1}{A}\sum_a min_k \mathbb{1}_{miss\,k}^a \tag{1}$$

For consistent scene multi-agent prediction, we report the joint scene-level metrics, where the average operation over the agents is done before the minimum operator. In this formulation, the minimum is taken over scene modalities, meaning that only the best scene (joint over agents) modality is taken into account:

$$minSFDE_k = min_k \frac{1}{A}\sum_a \|p_k^a - \hat{p}^a\|_2 \, , \; SMR_k = min_k \frac{1}{A}\sum_a \mathbb{1}_{miss\,k}^a \tag{2}$$

In other words, the marginal metrics pick their optimal solutions in a pool of $k$ to the power of $a$ predicted solutions, while the joint metrics restrict this pool to only $k$ possibilities, making it a much more complex problem.

We also report the scene collision rate (SCR) which is the percentage of modalities where two or more agents collide together, and the consistent collision-free joint Miss Rate (cSMR), which is $SMR$ where colliding modalities are also counted as misses.

### 4.3 COMPARISON WITH STATE-OF-THE-ART

We compare our THOMAS model performance with other joint predictions methods ILVM (Casas et al., 2020) and SceneTransformer (Ngiam et al., 2021). For fair comparison, we use a GOHOME encoder for each of the method, and adapt them accordingly so that they predict only endpoints similar to our method. For each method, we focus on implementing the key idea meant to solve scene consistency and keep the remaining part of the model as close as possible to our approach for fair comparison:

**ILVM (Casas et al., 2020)** uses variational inference to learn a latent representation of the scene conditioned of each agent with a Scene Interaction Module, and decodes it with a similar Scene Interaction Module. The required number of modalities is obtained by sampling the latent space as many times as require. Even though the sampling is independent for each agent, the latent space is generated conditionally on all the agents.

**SceneTransformer (Ngiam et al., 2021)** duplicates the agent encoding with the number of modalities required and adds a one-hot encoding specific to each modality. They then apply a shared transformer architecture on all modalities to encode intra-modality interactions between agents and generate the final trajectories.

More details on the state-of-the-art method and the re-implementations we used are given in Appendix A.4. The results are reported in Tab. 1. While having comparable marginal distance performance (demonstrating that our model is not inherently more powerful or leveraging more information), THOMAS significantly outperforms other methods on every joint metric. SMR is improved by about 25% and SCR by almost 30%, leading to a combined cSMR decreased by also more than 25%. We also report our numbers from the Interpret multi-agent online test set leaderboard in Tab.

Table 1: Comparison of consistent solutions on Interpret multi-agent validation track

|  | Marginal metrics | | | Joint metrics | | | |
|---|---|---|---|---|---|---|---|
|  | mADE | mFDE | MR | mFDE | MR | SCR | cMR |
| ILVM (Casas et al., 2020) | 0.30 | 0.62 | 10.8 | 0.84 | 19.8 | 5.7 | 21.3 |
| SceneTranformer (Ngiam et al., 2021) | **0.29** | **0.59** | 10.5 | 0.84 | 15.7 | 3.4 | 17.3 |
| THOMAS | 0.31 | 0.60 | **8.2** | **0.76** | **11.8** | **2.4** | **12.7** |

5 of Appendix A.5. For better comparison with existing single-agent trajectory prediction state-of-the-art, we provide evaluation numbers on the single-agents benchmarks Argoverse (Chang et al., 2019) and NuScenes (Caesar et al., 2020) in Appendix A.7.

## 4.4 ABLATION STUDIES

### 4.4.1 RECOMBINATION MODULE

We establish the following baselines to assess the effects our THOMAS recombination.

**Scalar output**: we train a model with the GOHOME graph encoder and a multimodal scalar regression head similar to Liang et al. (2020). We optimize it with either marginal and joint loss formulation.

**Heatmap output** with deterministic sampling: we try various sampling methods applied on the heatmap, with either the deterministic sampling as described in Gilles et al. (2021a) or a learned sampling trained to directly regress the sampled modalities from the input heatmap.

We report the comparison between the aforementioned baselines and THOMAS in Tab. 2. Compared to these baselines, THOMAS can be seen as an hybrid sampling method that takes the result of deterministic sampling as input and learns to recombine it into a more coherent solution. With regard to the joint algorithmic sampling that only tackled collisions but has little to no effect on actual consistency, as highlighted by the big drop in collision rate from 7.2% to 2.6% but a similar joint SMR, THOMAS actually brings a lot of consistency in the multi-agent prediction and drops the joint SMR from 14.8% to 11.8%.

Table 2: Comparison of consistent solutions on Interpret multi-agent validation track

| Output | Sampling | Objective | Marginal metrics | | | Joint metrics | | | |
|---|---|---|---|---|---|---|---|---|---|
|  |  |  | mADE | mFDE | MR | mFDE | MR | Col | cMR |
| Scalar | – | Marg | 0.28 | 0.59 | 10.4 | 1.04 | 23.7 | 6.4 | 24.9 |
| Scalar | – | Joint | 0.34 | 0.77 | 16.2 | 0.90 | 19.9 | 49 | 21.7 |
| Heat | Learned | Marg | **0.26** | **0.46** | 4.9 | 0.98 | 20.9 | 4.1 | 21.9 |
| Heat | Learned | Joint | 0.29 | 0.58 | 9.8 | 0.88 | 15.2 | 3.0 | 16.4 |
| Heat | Algo | Marg | 0.29 | 0.54 | **3.8** | 0.83 | 14.8 | 7.2 | 15.9 |
| Heat | Algo | Joint | 0.29 | 0.54 | **3.8** | 0.83 | 14.8 | 2.6 | 15.6 |
| Heat | Combi | Joint | 0.31 | 0.60 | 8.2 | **0.76** | **11.8** | **2.4** | **12.7** |

Usually, scalar marginal models already suffer from learning difficulties as only one output modality, the closest one to ground, can be trained at a time. Some modalities may therefore converge faster to acceptable solutions, and benefit from a much increased number of training samples compared to the others. This problem is aggravated in the joint training case, since the modality selected is the same for all agents in a training sample. The joint scalar model therefore actually fails to learn multi-modality as illustrated by a higher marginal $\text{minFDE}_6$ than any other model, and an extremely high crossCollisionRate since some modalities never train and always point to the same coordinates regardless of the queried agent. Note that, despite a similar training loss, SceneTransformer doesn't suffer of the same pitfalls in Tab. 1 as is shares the same weights between all modalities and only differentiates them in the initialization of the features.

## 4.5 HIERARCHICAL HEATMAP REFINEMENT

We assess the speed gain of our proposed hierarchical decoder compared to the lane rasters of GO-HOME Gilles et al. (2021a). In Tab. 3. We report training time for 16 epochs with a batchsize of 16, and inference time for 32 and 128 simultaneous agents.

Table 3: Comparison of consistent solutions on Interpret multi-agent validation track

|  | Training | Inference 32 agents | Inference 128 agents |
| --- | --- | --- | --- |
| GOHOME | 12.8 hours | 36 ms | 90 ms |
| THOMAS | 7.5 hours | 20 ms | 31 ms |

For additional comparison, the other existing dense prediction method DenseTNT (Gu et al., 2021) reports an inference speed of 100ms per sample for their model.

## 4.6 QUALITATIVE EXAMPLES

In this section, we will mostly compare the model before recombination, which we will refer by the $Before$ model, to the model after recombination, referenced as the $After$ model. We display four qualitative examples in Fig. 5 with colliding modalities in the $Before$ model (in dashed orange) and the solved modality (in full line orange) after recombination. For each model ($Before$-dashed or $After$-full), the highlighted modality in orange is the best modality according to the $SMR_6$ metric among the 6 available modalities. We also display in dashed grey the other 5 predicted $Before$ modalities, and highlight that the recombination model indeed selects modalities already available in the vanilla set and reorders them so that non-colliding modalities are aligned together.

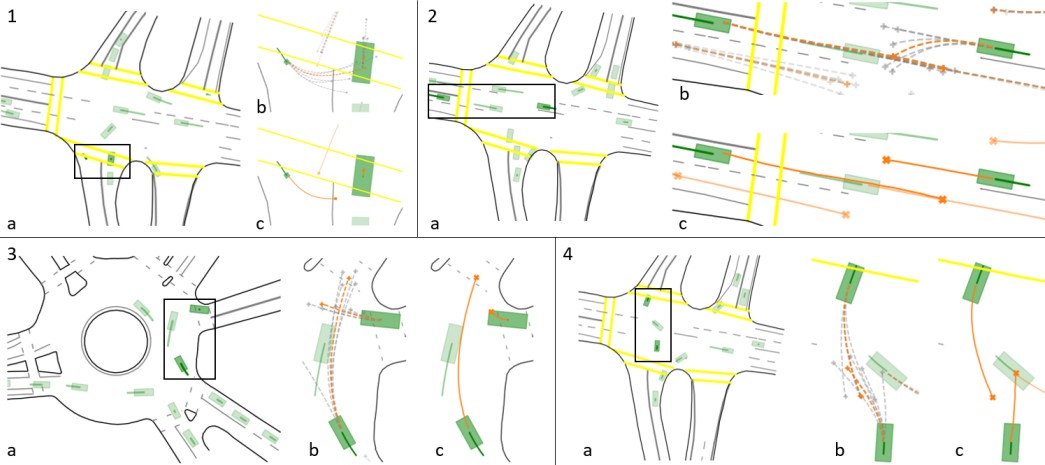

Figure 5: Qualitative examples of recombination model assembling collision-free modalities together compared to initial colliding modalities. For each example we display the general context with highlighted agents and area of interest, then two zooms in on the agents, one displaying the initial best modality before recombination in dashed orange and all the other available modalities in grey. The second zooms shows the best modality after recombination in full line orange.

We also show more qualitative examples in Fig. 6, where we highlight the comparison in modality diversity between the $Before$ model (in dashed lines) and the $After$ model (in full lines). While the $Before$ model tries to spread the modalities for all agents to minimize marginal miss-rate, the recombined model presents much less spread compared to the original model, maintaining a multimodal behavior only in presence of very different possible agent intentions such as different possible exits or turn choices. For most other agents, almost all modalities are located at the same position, that is, the one deemed the most likely by the model. Thus, if the truly uncertain agents

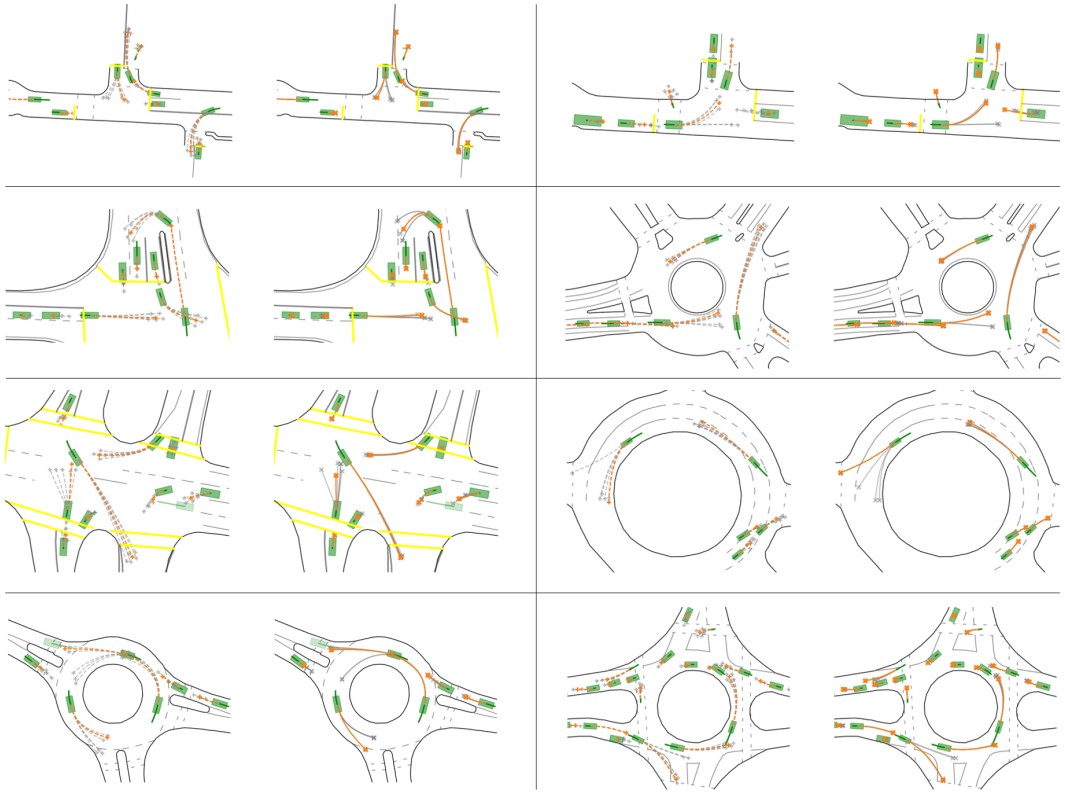

Figure 6: Qualitative examples of recombination model selecting fewer but more pronounced and impactful agent modalities compared to initial colliding modalities. For each example we display on the left the vanilla model modalities with dashed lines, with the initial best modality in dashed orange and all the other available modalities in grey. On the right we display the selected modalities after recombination, where the model focuses on the most likely occurrence in most agents.

have to select the second or third most likely modality, the other agents still have their own most likely modality

## 5   CONCLUSION

We have presented THOMAS, a recombination module that can be added after any trajectory prediction module outputting multi-modal predictions. By design, THOMAS allows to generate scene-consistent modalities across all agents by making the scene modalities select coherent agent modalities and restricting the modality budget on the agents that truly need it. We show significant performance increase when adding the THOMAS module compared to the vanilla model and achieve state-of-the-art results compared to already existing methods tackling scene-consistent predictions.
.

REPRODUCIBILITY STATEMENT

We use the publicly available Interaction 1.2 dataset (Zhan et al., 2019) available at `http://challenge.interaction-dataset.com/dataset/download`. We detail dataset preprocessing in Appendix A.2, training process in Appendix A.3 and give a precise illustration of our model architecture with each layer size in A.8.

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

# A APPENDIX

## A.1 METRIC DETAILS

We adopt for each dataset evaluation the miss definition as specified by the dataset benchmark. Argoverse (Chang et al., 2019) and NuScenes (Caesar et al., 2020) define a miss as the prediction being further than 2 meters to the ground truth, while Waymo (Ettinger et al., 2021) and Interaction Zhan et al. (2019)count a miss when the prediction is closer than a lateral (1m) and an longitudinal threshold with regard to speed:

$$\text{Threshold}_{\text{lon}} = \begin{cases} 1 & v < 1.4m/s \\ 1 + \frac{v-1.4}{11-1.4} & 1.4m/s \le v \le 11m/s \\ 2 & v \ge 11m/s \end{cases}$$

## A.2 DATASET PROCESSING

We use the training/validation split provided in Interaction 1.2. We select for each scene a reference agent on which the scene will be centered and oriented according to its heading at prediction time. At training time, this reference agent is chosen randomly across all agents. At validation and testing time, we compute the barycenter of all agents and select the closest agent as the scene reference point. We use for training all agents that have ground-truth available at prediction horizon T=3s.

We use the provided dataset HD-Maps to create the context graphs. As in Liang et al. (2020), we use four relations:$\{predecessor, successor, left, right\}$ obtained from the lane graph connectivity. We upsample long lanelets to have a maximum of 10 points in each lanelet.

For each agent, we use their trajectory history made of position, yaw and speed in the past second sampled at 10Hz. We also provide the model a mask indicating if the agent was present in a given time-frame, and pad with zeros when the agent is not tracked at a specific timestep.

## A.3 TRAINING DETAILS

We train all models with Adam optimizer and batchsize 32. We initialize the learning rate at $1e^{-3}$ and divide it by 2 at epochs 3, 6, 9 and 13, before stopping the training at epoch 16. We use ReLU activation after every linear layer unless specified otherwise, and LayerNormalization after every attention and graph convolution layer.

During training, since a scene can contain up to 40 agents, which is too much for gradient computation in batches of 32, we restrict the number of predicted agents to 8 by randomly sampling them across available agents. We do not use other data augmentation.

The final heatmap $Y$ predicted cover a range of 192 meters at resolution 0.5m, hence a (384, 384) image. For heatmap generation, we use the same pixel-wise focal loss over the pixels $p$ as in Gilles et al. (2021b):

$$L = -\frac{1}{P} \sum_p (Y_p - \hat{Y}_p)^2 f(Y_p, \hat{Y}_p) \text{ with } f(Y_p, \hat{Y}_p) = \begin{cases} \log(\hat{Y}_p) & \text{if } Y_p = 1 \\ (1 - Y_p)^4 \log(1 - \hat{Y}_p) & \text{else} \end{cases}$$

where $\hat{Y}$ is defined as a Gaussian centered on the agent future position at prediction horizon $T$ with a standard deviation of 4 pixels, equivalent to 2m.

## A.4 BASELINES IMPLEMENTATIONS

### A.4.1 IMPLICIT LATENT VARIABLE MODEL

We use a GOHOME encoder for the prior, posterior and decoder Scene Interaction Modules. We weight the KL term with $\beta = 1$ which worked best according to our experiments.

### A.4.2 SCENE TRANSFORMER

The initial paper applies a transformer architecture on a $[F, A, T, D]$ tensor where $F$ is the potential modality dimension, $A$ the agent dimension and $T$ the time dimension, with $D$ the feature embedding, with factorized self-attention to the agent and time dimensions separately, so that agents can look at each-other inside a specific scene modality. The resulting output is optimized using a jointly formalized loss. For our implementation, we get rid of the $T$ dimension as we focus on endpoint prediction and coherence between the $A$ agents. The initial encoded $[A, D]$ tensor is obtained with a GOHOME encoder, multiplied across the $F$ futures and concatenated with a modality-specific one-hot encoding as in Ngiam et al. (2021) to obtain the $[F, A, D]$ tensor. We then apply two layers of agent self-attention similar to the original paper, before decoding the endpoints through a MLP.

### A.4.3 COLLISION-FREE ENDPOINT SAMPLING BASELINE

We designed a deterministic sampling algorithm based on the heatmaps generated in previous section in order to sample endpoints for each agent in a collision aware manner. We use the same sampling algorithm as Gilles et al. (2021a) based on MR optimization, but add a sequential iteration over the agents for each modalities.

For a single modality $k$, we predict the possible endpoint of a first agent $a$ by taking the maximum accumulated predicted probability under an area of radius $r$. We then not only set to zero the heatmap values of this agent heatmap $\mathcal{I}_{k'}^{a}$ around the sampled location so not to sample it in the next modalities $k'$, but we also set to zero the same area on the heatmaps $\mathcal{I}_{k}^{a'}$ of the other agents $a'$ on the same modality $k$, so that these other agents cannot be sampled at the same position for this modality. This way, we try to enforce collision-free endpoints, and expect that considering collisions brings logic to improve the overall consistency of the predictions. However, as will be highlighted in Sec. 4.4, this methods significantly improves the collision rate without the need for any additional learned model but it does barely improve the multi agent consistency.

### A.5 RESULTS ON INTERPRET CHALLENGE

We also report our numbers from the Interpret multi-agent track challenge online leaderboard in Table 5. It is to be noted that the DenseTNT solution explicitly checks for collisions in the search for its proposed predictions, which we don't, hence their $0\%$ collision rate (SCR) and its direct impact on consistent collision-free joint Miss Rate (cSMR).

Table 4: Results on Interpret multi-agent regular scene leaderboard (test set)

|  | minSADE | minSFDE | SMR | SCR | cSMR |
|---|---|---|---|---|---|
| MoliNet | 0.73 | 2.55 | 44.4 | 7.5 | 47.4 |
| ReCoG2 (Mo et al., 2020) | 0.47 | 1.16 | 23.8 | 6.9 | 26.8 |
| DenseTNT (Gu et al., 2021) | 0.42 | 1.13 | 22.4 | **0.0** | **22.4** |
| THOMAS | **0.42** | **0.97** | **17.9** | 12.8 | 25.2 |

Table 5: Results on Interpret multi-agent conditional scene leaderboard (test set)

|  | minSADE | minSFDE | SMR | SCR | cSMR |
|---|---|---|---|---|---|
| ReCoG2 (Mo et al., 2020) | 0.33 | 0.87 | 14.98 | 0.09 | 15.12 |
| DenseTNT (Gu et al., 2021) | 0.28 | 0.89 | 15.02 | **0.0** | 15.02 |
| THOMAS | **0.31** | **0.72** | **10.67** | 0.84 | **11.63** |

### A.6 SPEED / PERFORMANCE TRADE-OFF WITH HIERARCHICAL REFINEMENT

We also display the trade-off between inference speed and coverage from hierarchical refinement in Fig. 7, evaluated on the Interpret multi-agent dataset with marginal MissRate$_6$. The curve is obtained setting the number $N$ of upsampled points at the last refinement iteration from 2 to 128.

From $N = 16$ and lower, coverage performance starts to diminish while little speed gains are made. We still kept a relatively high N=64 in our model as we wanted to insure a wide coverage, and the time loss between 41 ms and 46 ms remains acceptable.

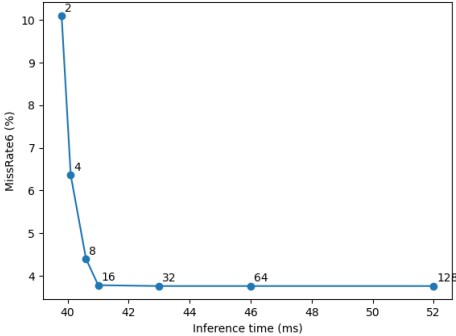

Figure 7: Curve of MissRate$_6$ with regard to inference time with varying number of points upsampled at the last hierarchical refinement iteration.

## A.7 ADDITIONAL RESULTS ON SINGLE-AGENT DATASETS

Table 6: Argoverse Leaderboard (Chang et al.)

| | K=1 | | K=6 | | |
|---|---|---|---|---|---|
| | minFDE | MR | minADE | minFDE | MR |
| LaneGCN (Liang et al., 2020) | 3.78 | 59.1 | 0.87 | 1.36 | 16.3 |
| Autobot (Girgis et al., 2021) | – | – | 0.89 | 1.41 | 16 |
| TPCN (Ye et al., 2021) | 3.64 | 58.6 | 0.85 | 1.35 | 15.9 |
| Jean (Mercat et al., 2020) | 4.24 | 68.6 | 1.00 | 1.42 | 13.1 |
| SceneTrans (Ngiam et al., 2021) | 4.06 | 59.2 | **0.80** | **1.23** | 12.6 |
| LaneRCNN (Zeng et al., 2021) | 3.69 | 56.9 | 0.90 | 1.45 | 12.3 |
| PRIME (Song et al., 2021) | 3.82 | 58.7 | 1.22 | 1.56 | 11.5 |
| DenseTNT (Gu et al., 2021) | 3.70 | 59.9 | 0.94 | 1.49 | 10.5 |
| GOHOME (Gilles et al., 2021a) | 3.65 | 57.2 | 0.94 | 1.45 | 10.5 |
| HOME (Gilles et al., 2021b) | 3.65 | 57.1 | 0.93 | 1.44 | **9.8** |
| THOMAS | **3.59** | **56.1** | 0.94 | 1.44 | 10.4 |

Table 7: NuScenes Leaderboard (Caesar et al.)

| | K=5 | | K=10 | | k=1 |
|---|---|---|---|---|---|
| | minADE | MR | minADE | MR | minFDE |
| CoverNet (Phan-Minh et al., 2020) | 1.96 | 67 | 1.48 | – | – |
| Trajectron++ (Salzmann et al., 2020) | 1.88 | 70 | 1.51 | 57 | 9.52 |
| ALAN (Narayanan et al., 2021) | 1.87 | 60 | 1.22 | 49 | 9.98 |
| SG-Net (Wang et al., 2021) | 1.86 | 67 | 1.40 | 52 | 9.25 |
| WIMP (Khandelwal et al., 2020) | 1.84 | 55 | 1.11 | 43 | 8.49 |
| MHA-JAM (Messaoud et al., 2020) | 1.81 | 59 | 1.24 | 46 | 8.57 |
| CXX (Luo et al., 2020) | 1.63 | 69 | 1.29 | 60 | 8.86 |
| LaPred (Kim et al., 2021) | 1.53 | – | 1.12 | – | 8.12 |
| P2T (Deo & Trivedi, 2020) | 1.45 | 64 | 1.16 | 46 | 10.50 |
| GOHOME (Gilles et al., 2021a) | 1.42 | 57 | 1.15 | 47 | 6.99 |
| Autobot (Girgis et al., 2021) | 1.37 | 62 | 1.03 | 44 | 8.19 |
| PGP (Deo & Trivedi, 2020) | **1.30** | 57 | **0.98** | **37** | 7.72 |
| THOMAS | 1.33 | 55 | 1.04 | 1.04 | **6.71** |

## A.8 DETAILED ARCHITECTURE

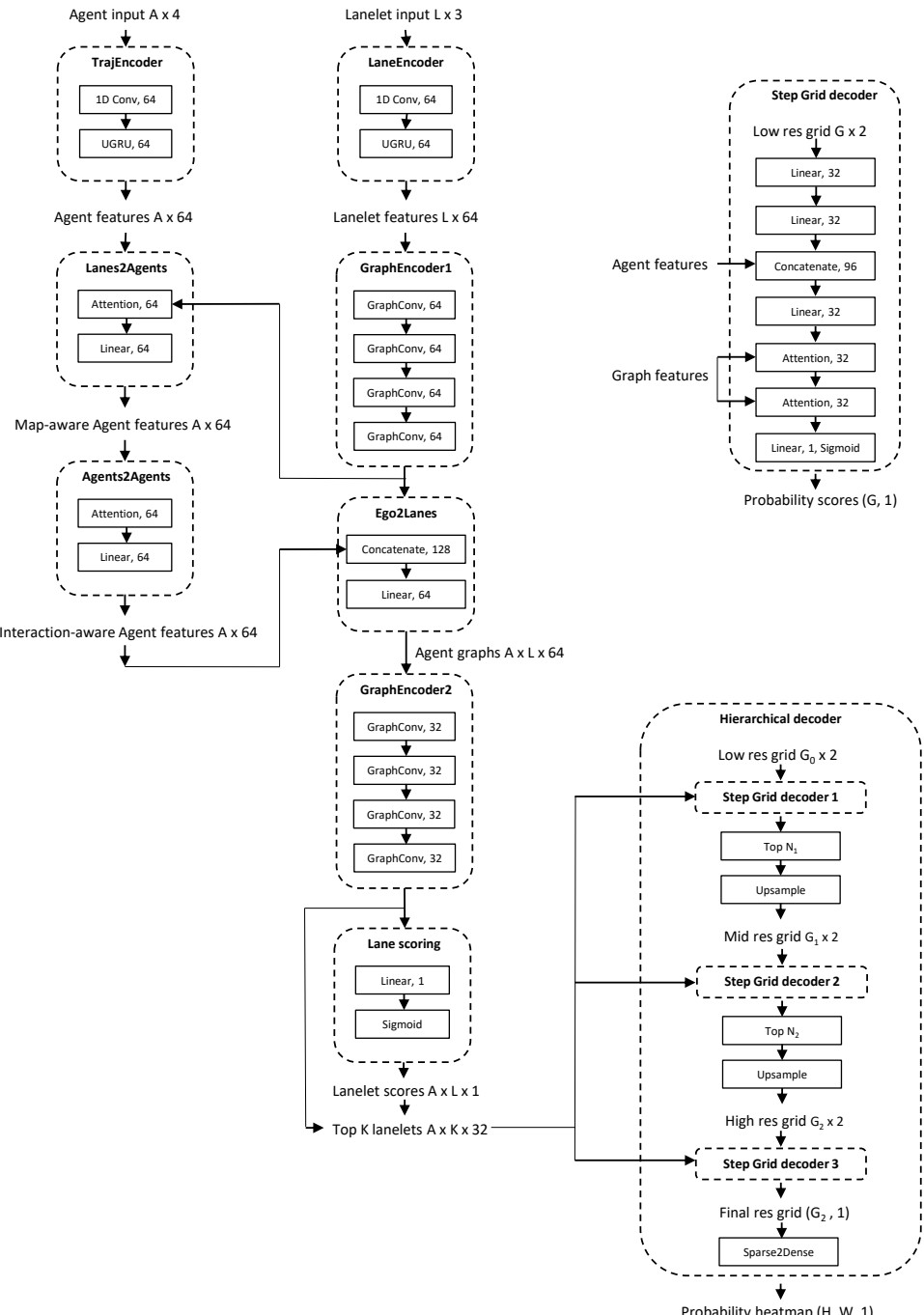

Figure 8: Detailed illustration of our heatmap generator model

A.9 HEATMAP QUALITATIVE RESULTS

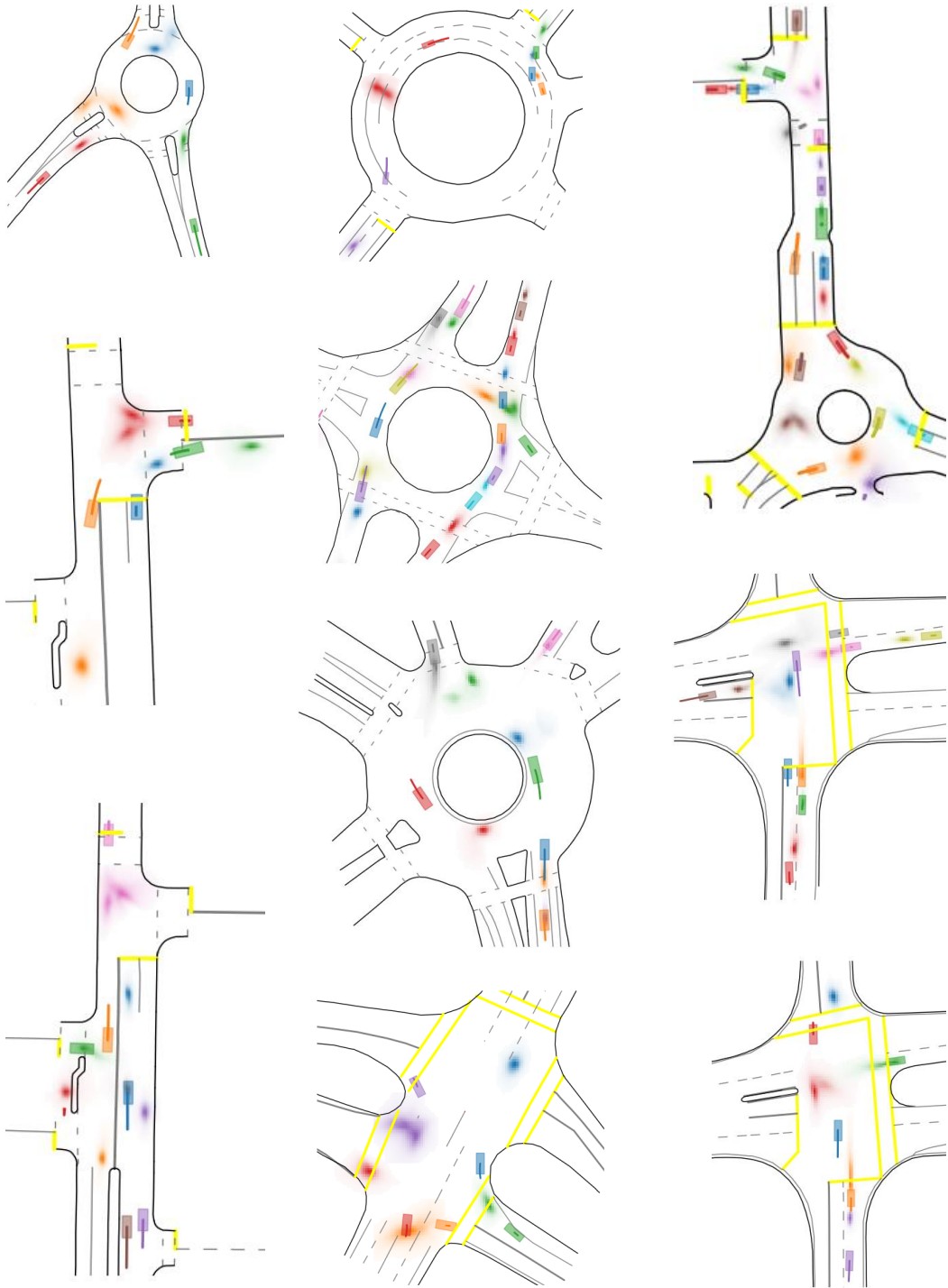

Figure 9: Qualitative examples of heatmap output from our multi-agent model. All the heatmaps from one scene are the results from one single forward pass in our model predicting all agents at once. We use matching colors for the agent history, current location and their future predicted heatmap (best viewed in color).

