# OpenReview forum: "THOMAS: Trajectory Heatmap Output with learned Multi-Agent Sampling"
_ICLR.cc/2022/Conference — ICLR 2022 Poster_

### Official Review · Reviewer_ZJY1 · 2021-10-31

**Correctness:** 3
**Technical Novelty And Significance:** 3
**Empirical Novelty And Significance:** 2
**Recommendation:** 6
**Confidence:** 5

**Main Review:**

### Strengths:
- __Originality__: the two main contributions outlined in the summary of the paper are fairly distinctive from previous works.
  - The hierarchical heatmap prediction tackles one of the main issues of heatmap representations, which is their heavy memory consumption.
  - The learnable recombination method is general enough to be suitable for any marginal prediction model, which makes me think it would have a solid impact on the community.
- Introduction and related work are succinct and touch upon the main relevant points to the future prediction task.
- Figures are useful to understand the method as well as to evaluate the performance qualitatively.

### Weaknesses:
- __Writing ambiguity and lack of details__: I am not sure the write-up is polished enough for submission to ICLR. I outline a few reasons below:
  - Throughout the paper, there is no mention or description of the loss functions utilized, and barely any mention of the overall training process (with the exception of a sentence in 3.1.3). In the appendix, the authors describe how the heatmap output is optimized, but there is no mention about how the learnable recombination module is trained, which is one of the main contributions of the paper.
  - Because of this lack of description of the overall training process and loss functions, it is unclear if in the experiments the authors utilize 2 separate THOMAS models - one for marginal prediction and one for joint prediction. For instance, are the marginal metrics and joint metrics in Table 1 coming from a single model, or two separately trained models? A sentence that contributes to this confusion is in 3.1.3, where the authors mention that they decode K end points using the same MR optimization algorithm as Gilles et al. (2021b). However, in 3.2.1 the authors enhance this algorithm with collision-free sampling. Please clarify this in the rebuttal.
  - The paper criticizes generative models in the second paragraph of the related work for not providing a probability value for each prediction, which makes me expect the method to provide such scores. However, there is no mention about it in the approach or experiments sections.
  - In Table 3, the Marg and Joint objectives are not defined.
- It is unclear how some pieces tie together. My main concern is with respect to collision-free end point sampling (3.2.1) and recombination (3.2.2). The heuristic in 3.2.1 seems to assume that the modalities are "already ordered" since the algorithm basically avoids sampling close end points across actors for a given modality k. However, the recombination can perform arbitrary linear combinations across modes, thus not respecting this. Is the idea that only one of the two methods are used to perform inference/training on a model? If so, I would suggest moving 3.2.1 to the experiments as a baseline/ablation, and I definitely would not claim it as a contribution (particularly since it's just a small tweak on top of a previously proposed algorithm).
- Experiments are not as thorough as I would expect. This is understandable for the joint prediction setting as there are very few baselines and the dataset/benchmarks on this task have just started. In fact, I appreciate the effort of the authors of re-implementing ILVM and SceneTransformer. However, with respect to marginal prediction, there are a lot of methods and benchmarks out there. Another aspect that would have been great to get more insight on is the hierarchical heatmap output. What's the tradeoff between computation (both runtime and memory) and accuracy, with respect to dense heatmap approaches? How was the current 3-layer hierarchy chosen?

### Additional feedback:
In this section I provide additional feedback for rebuttal, but please address first the main points above.
- Re: training details in the appendix. I find it surprising that the method is trained with random sampling of up to 8 agents, given that for joint prediction there might be other agents that are particularly relevant to the task. Was this better than lowering the batch size? If so, is it because it is rare that a scene contains > 8 actors in the INTERACTION dataset?
- The equation that describes the weighted linear combination of agent modalities seems incomplete. As far as I understand, there should be a weighted sum in the equation, using the scores coming out of the softmax.
- Related work is missing some relevant works. In the last paragraph, joint autoregressive methods should be mentioned (e.g., PRECOG[A] - see references at the bottom) as they were the first to model the interactions across agents at the output level. With respect to generative models and their random sampling and lack of confidence scores per modality, LookOut[B] is a recent method that addresses both these concerns with a learnable diverse sampler and scene scoring. The authors can treat this as concurrent work since it was just published at ICCV, but please include it in the discussion as it is highly related.
- Is the notation of the joint metrics coming from the Interpret challenge? If not, I would suggest unifying this with the names proposed originally in ILVM (e.g., minJointFDE --> minSFDE, CrossCollisionRate --> SCR).
- While the sparse and hierarchical heatmaps make sense to mitigate their high resource consumption, I am a bit concerned about the fact that endpoints are only sampled from the finest resolution, as this might not contain more unlikely but safety-critical modalities. Overall, the heuristic refinement of the most likely grid cells seems contradictory with the endpoint sampling objective of maximizing coverage.
- The writing would benefit from more intuitions. For instance, the method samples heuristically from the heatmaps to maximize coverage, but why is this important to the overall self-driving task?
- I personally did not get much out of figure 4, perhaps adding the shapes of the tensors could be helpful.
- I would avoid calling the combination method an "ordering", as the method is more general than that.
- How come the CrossCollisionRate is still 2.6% when using the collision-free end point sampling? I would expect it to be zero.

### Additional references:
- [A] Rhinehart, Nicholas, et al. "Precog: Prediction conditioned on goals in visual multi-agent settings." Proceedings of the IEEE/CVF International Conference on Computer Vision. 2019.
- [B] Cui, Alexander, et al. "LookOut: Diverse multi-future prediction and planning for self-driving." Proceedings of the IEEE/CVF International Conference on Computer Vision. 2021.

**Summary Of The Paper:**

The paper proposes a joint trajectory prediction framework. This means that the model produces multi-modal future predictions at the scene level (i.e., for all actors jointly). There has only been very recent research in this area despite its importance to autonomy and motion planning in particular, so the relevance of the paper to the broader motion forecasting field is high. I identify two main contributions in the paper: (i) hierarchical and sparse heatmap output representation and (ii) a recombination method that is pluggable in any multi-modal marginal prediction model.

**Summary Of The Review:**

I think this paper has significant contributions in terms of the presented method. However, my main concerns are related to the lack of details in the writing as well as the thoroughness of the experiments and the insights that can be extracted from them. For these reasons, I am leaning towards rejection, but I look forward to the authors' response during the rebuttal period.

### Post-rebuttal

After careful consideration of the rebuttal, I am happy to increase my score and I lean towards acceptance. My doubts were clarified, and the authors have timely adapted the write-up. Since my main issue was with the lack of clarity in some parts and that has been improved I raise my score to a 6. I am in between 6 and 8, but I think 8 is a bit too much since the improvements are incremental. I would rate it as a 7 if I had the option given that the proposed linear recombination of marginal predictions is very flexible and could be easily adapted to multiple prior works in the literature, thus raising the future impact of this work.

---

> ### Author Response · Authors · 2021-11-16
> **Response to Reviewer ZJY1 (Part 1)**
>
> Thanks a lot for your detailed and valuable feedback !
>
> > Throughout the paper, there is no mention or description of the loss functions utilized, and barely any mention of the overall training process (with the exception of a sentence in 3.1.3). In the appendix, the authors describe how the heatmap output is optimized, but there is no mention about how the learnable recombination module is trained, which is one of the main contributions of the paper.
> Because of this lack of description of the overall training process and loss functions, it is unclear if in the experiments the authors utilize 2 separate THOMAS models - one for marginal prediction and one for joint prediction. For instance, are the marginal metrics and joint metrics in Table 1 coming from a single model, or two separately trained models? A sentence that contributes to this confusion is in 3.1.3, where the authors mention that they decode K end points using the same MR optimization algorithm as Gilles et al. (2021b). However, in 3.2.1 the authors enhance this algorithm with collision-free sampling. Please clarify this in the rebuttal.
>
> We apologize for the lack of clarity. The loss used to train the recombination module is actually the joint formulation of minJointFDE:
>
> $$minJointFDE_k = min_k \frac{1}{A} \sum_a  \rVert p^a_k-\hat{p}^a\rVert_2 $$
>
> Similarly for the scalar and learned sampling baselines in Tab. 3, the marginal objective is the marginal definition of minFDE:
>
> $$minFDE_k = \frac{1}{A} \sum_a min_k \rVert p^a_k-\hat{p}^a\rVert_2 $$
>
> And the joint objective in Tab. 3 is again the joint formulation of minJointFDE.
> Whether it is in Tab. 1 or Tab. 3, one row always corresponds to a single training of a single model. In Tab. 1, since the focus of the paper is joint multi-agent prediction, all models have been trained with the joint loss specified above (with the exception of ILVM [1] which is trained with their own variational learning objective specified in their paper). We included marginal results for better comparison with previous work, as well as highlighting that our method doesn’t necessarily leverage more innate computing power.
> We will try to explain this better in the revised version of the paper.
>
> >The paper criticizes generative models in the second paragraph of the related work for not providing a probability value for each prediction, which makes me expect the method to provide such scores. However, there is no mention about it in the approach or experiments sections.
>
> The initial sampling algorithm from [2] provides probabilities as accumulated heatmap values under the sampled area for each sampled endpoint. Additionaly, we indeed overlooked to specify that the recombination module predicts a probability value from each scene modality features. This probability value is trained to be 1 for the closest modality according to jointFDE and 0 otherwise using a binary cross-entropy loss.
>
> >It is unclear how some pieces tie together. My main concern is with respect to collision-free end point sampling (3.2.1) and recombination (3.2.2). The heuristic in 3.2.1 seems to assume that the modalities are "already ordered" since the algorithm basically avoids sampling close end points across actors for a given modality k. However, the recombination can perform arbitrary linear combinations across modes, thus not respecting this. Is the idea that only one of the two methods are used to perform inference/training on a model? If so, I would suggest moving 3.2.1 to the experiments as a baseline/ablation, and I definitely would not claim it as a contribution (particularly since it's just a small tweak on top of a previously proposed algorithm).
>
> Indeed 3.2.1 was another approach to consider multi-agent consistent sampling, and we definitely agree with the reviewers to move it as a baseline/ablation for clarity.
>
> >Experiments are not as thorough as I would expect. This is understandable for the joint prediction setting as there are very few baselines and the dataset/benchmarks on this task have just started. In fact, I appreciate the effort of the authors of re-implementing ILVM and SceneTransformer. However, with respect to marginal prediction, there are a lot of methods and benchmarks out there.
>
> Following the reviews, we are currently working to evaluate our model on the Argoverse and NuSCenes benchmarks, but indeed as you mentioned these only work for marginal prediction. We are therefore also working on adapting NuScenes for the multi-agent case and providing evaluation of our model on this framework as well.
>
> Preliminary results on the validation sets of Argoverse and NuScenes give the following numbers (official submission on online leaderboard remains to be done):
>
> | Argoverse validation| minFDE6 |  MR6 |
> | -----------| ----|---|
> | GOHOME  | 1.26    |7.1  |
> | THOMAS | 1.22    |6.6  |
>
> | NuScenes validation| minADE5  |  MR5 | minFDE1 |
> | --------| ----|----|----|
> | GOHOME | 1.42  | 57   |6.99 |
> | THOMAS   | 1.34  | 54   |6.73  |

---

> ### Author Response · Authors · 2021-11-16
> **Response to Reviewer ZJY1 (Part 2)**
>
> >Another aspect that would have been great to get more insight on is the hierarchical heatmap output. What's the tradeoff between computation (both runtime and memory) and accuracy, with respect to dense heatmap approaches? How was the current 3-layer hierarchy chosen?
>
> We are currently running and trade-off experiments so that we could enrich the paper with these numbers by the end of the rebuttal.
> Preliminary results on training and inference time on a single 2080 RTX GPU give the following numbers on the Interpret dataset with a batchsize of 16 (training lasts 16 epochs):
>
> | Computational cost| Inference time  |  Training time |
> | --------- | --------|----- |
> | GOHOME | 33 ms   | 12.8 hours   |
> | THOMAS | 21 ms    | 7.5 hours  |
>
> (Note that these numbers are on the single agent track, and don't account for the decoding part having to be repeated multiple times in the multi-agent case. We will try to provide additional comparison for the multi-agent case.)
>
> We are working to get more experiments highlighting the trade-off and justifying our choices of parameters for the hierarchical heatmap output.
>
> >Re: training details in the appendix. I find it surprising that the method is trained with random sampling of up to 8 agents, given that for joint prediction there might be other agents that are particularly relevant to the task. Was this better than lowering the batch size? If so, is it because it is rare that a scene contains > 8 actors in the INTERACTION dataset?
>
> We apply a distinction between the number of agents taken as input, which can be far greater than 8 and isn't restricted, and the number of predicted agents. During training, a predicted agent needs to have all 3 seconds in the future available, and this only happens for a restricted subset of agents in the scene, which we found to be on average close to 8 in most scenarios. We indeed found it more beneficial to restrict the number of predicted agents at training time rather than lowering the batch size, as cases with more predictable agents rarely happened. However, while we only predict and back-propagate the loss for these 8 selected agents we still take all agents into account for interactions in the encoding module of the model.
>
> >The equation that describes the weighted linear combination of agent modalities seems incomplete. As far as I understand, there should be a weighted sum in the equation, using the scores coming out of the softmax.
>
> Indeed the extended equation is :
> $$ p_{l}^{a} = \sum_k softmax(s_{l}^{k_a}) p_k^a $$
> We intented to write it as the matrix product of $softmax(s_{l}^{k_a})$ and $p_k^a$ for brevity but missed a transpose operator and it seems to be less clear that way.
>
> >Related work is missing some relevant works. In the last paragraph, joint autoregressive methods should be mentioned (e.g., PRECOG[A] - see references at the bottom) as they were the first to model the interactions across agents at the output level. With respect to generative models and their random sampling and lack of confidence scores per modality, LookOut[B] is a recent method that addresses both these concerns with a learnable diverse sampler and scene scoring. The authors can treat this as concurrent work since it was just published at ICCV, but please include it in the discussion as it is highly related.
>
> Indeed we forgot to mention Precog [3] in the relative work. We didn’t include it as a baseline since ILVM [1] already compared positively to them and mentionned that they had to modify these kind of social auto-regressive methods to achieve competitive results. We will also include LookOut [4] in the revised related work, as it has a very interesting approach but seems to focus more on the planning aspects and end-to-end driving in closed-loop simulation rather than the prediction aspect, even the latter is still covered, the few prediction metric they report (minSADE in Fig5, minSADE in Tab. 4) are similar to ILVM.
>
> >Is the notation of the joint metrics coming from the Interpret challenge? If not, I would suggest unifying this with the names proposed originally in ILVM (e.g., minJointFDE --> minSFDE, CrossCollisionRate --> SCR).
>
> The notation is from the challenge (https://github.com/interaction-dataset/INTERPRET_challenge_multi-agent), but for clarity and brevity we will change our notation to the ones proposed originally in ILVM as suggested.

---

> ### Author Response · Authors · 2021-11-16
> **Response to Reviewer ZJY1 (Part 3)**
>
> >While the sparse and hierarchical heatmaps make sense to mitigate their high resource consumption, I am a bit concerned about the fact that endpoints are only sampled from the finest resolution, as this might not contain more unlikely but safety-critical modalities. Overall, the heuristic refinement of the most likely grid cells seems contradictory with the endpoint sampling objective of maximizing coverage.
>
> We are working on additional experiments on the coverage at latest resolution to justify our choices, in short the idea is that the number of refined points at the latest resolution is sufficient to cover all the future possibilities.
>
> >How come the CrossCollisionRate is still 2.6% when using the collision-free end point sampling? I would expect it to be zero.
>
> The collision-free end point sampling only prevents collisions at the last timestep (the one for the sampled endpoint), however collision checking is applied on all the future timesteps and collision can still happen in the middle of the trajectory.
>
> [1] Casas, Sergio, et al. "Implicit latent variable model for scene-consistent motion forecasting." In *ECCV*,  2020
>
> [2] Gilles, Thomas, et al. "HOME: Heatmap output for future motion estimation" In *ITSC*, 2021.
>
> [3] Rhinehart, Nicholas, et al. "Precog: Prediction conditioned on goals in visual multi-agent settings." In *CVPR*, 2019.
>
> [4] Cui, Alexander, et al. "LookOut: Diverse multi-future prediction and planning for self-driving." In *CVPR*,  2021.

---

### Official Review · Reviewer_PgCY · 2021-11-01

**Correctness:** 3
**Technical Novelty And Significance:** 2
**Empirical Novelty And Significance:** 2
**Recommendation:** 6
**Confidence:** 4

**Main Review:**


### Strong Points

The idea of performing re-estimation based on the computed heat map end-points
seems novel to me. This idea seems like a natural next step to improve multi agent motion
prediction. The fact that you first get the end-point and then reason about the scene is
intuitively sounding and seems to be a good solution for the problem.

The experiments on interact are a nice direction to be taken by the field.

The visualizations of the heat-maps and the different predictions was helpful.

### Weak Points

#### I could not find a clear distinction with literature
While reading the related work section I found hard to contrast the ideas of the paper with the ones proposed on the literature.
The difference between this method and GOHOME was  not clear to me in this sense.
From what I understood the main contribution here  is the modality combination based on the collision
free end-point sampling. However, this was not obvious for me when reading the paper.


#### The implemented baselines seemed to have performed way worse than expected.
For example, both [1] and [7] are methods designed for *joint* prediction and scene level prediction. This is contradicted
by the observations made by the authors on page 8, quote: " ... This problem is aggravated in the joint training case, since the modality selected is
the same for all agents in a training sample....". Both [1] and [3] fairly confidently claim producing multimodal joint predictions.
Either the version implemented by the authors does not correctly match the other works in the literature or there are specific findings
on the limitations existent in other methods.
Also, the paper lack comparison with latent variable based [2][5][[6] multi modal prediction methods. They seem to perform
better when the issue is diverse prediction. Definitely the mode does not tend to be the same for all predictions in those cases.


#### A single dataset evalution might not be enough for the community to access the value of this paper.

 Motion prediction is currently a very competitive scene. Even though I appreciate the idea on focusing reporting results on multi-agents
and joint prediction tasks, it is important to also evaluate the prediction on more standard benchmarks such as argoverse.
With results on argoverse, me and the rest of the community can more quickly access the the capabilities of the proposed
 heat map prediction pipeline, on producing consistent ego-agent motion prediction. I understand that this would not be the main
results, but it would be helpful.

##### Computational cost is apparently not shown.

I could not find the computational cost of running and training the proposed method on the paper. It seems that even
with the optimizations made on the heat map estimation the training time and inference time should be high.
Those kinds of motion prediction models have the necessity to further scale with way more data. Thus, highly
complex methods can be hard to train. There is also the case of potential future execution in an embedded hardware.

### Specific Questions

I am not  sure about the scene modality vectors ( Section 3.2.2), are they like seed parameters which are going to be used
to incorporate the joint consistent scene level ? I couldn't really find any sort of detailed explanation on that.

Why the lack of ADE for scene level metrics or using similar metrics as [3] ?


[1] Jiquan Ngiam, Benjamin Caine, Vijay Vasudevan, Zhengdong Zhang, Hao-Tien Lewis Chiang,
Jeffrey Ling, Rebecca Roelofs, Alex Bewley, Chenxi Liu, Ashish Venugopal, et al. Scene
transformer: A unified multi-task model for behavior prediction and planning. arXiv preprint
arXiv:2106.08417, 2021
[2] Tang, Charlie, and Russ R. Salakhutdinov. "Multiple futures prediction." Advances in Neural Information Processing Systems 32 (2019): 15424-15434.
[3] Li, Lingyun Luke, et al. "End-to-end contextual perception and prediction with interaction transformer." 2020 IEEE/RSJ International Conference on Intelligent Robots and Systems (IROS). IEEE, 2020.
[4] Yuan, Ye, et al. "AgentFormer: Agent-Aware Transformers for Socio-Temporal Multi-Agent Forecasting." arXiv preprint arXiv:2103.14023 (2021).
[5] Girgis, Roger, et al. "Autobots: Latent Variable Sequential Set Transformers." arXiv preprint arXiv:2104.00563 (2021).
[6] Casas, Sergio, et al. "Implicit latent variable model for scene-consistent motion forecasting." Computer Vision–ECCV 2020: 16th European Conference, Glasgow, UK, August 23–28,
[7] Ming Liang, Bin Yang, Rui Hu, Yun Chen, Renjie Liao, Song Feng, and Raquel Urtasun. Learning
lane graph representations for motion forecasting. In ECCV, 2020


**Summary Of The Paper:**

The paper proposes a multi-modal trajectory prediction pipeline.
The idea is that based on past trajectories and a map of the road a model
is trained to predict a heat map with the most likely end-points.
With those end-points a trajectory can be inferred for each agent in a collision
free way by leveraging all heat  maps jointly.
With those trajectories, multiple, consistent scene level trajectories are predicted.

The main contribution of the paper is on the reasoning about the end-points
given by the predicted heat map.

**Summary Of The Review:**


The paper provides an incremental solution to heatmap based motion
predictions that improves on scene level predictions.
Even though this idea is interesting, the experiments were not able
to convince me as this being an impactful approach for scene level prediction.

# Post rebuttal review

After carefully reading the authors rebuttal, I decided to increase my score.
The changes proposed by the authors and the explanations provided would increase the quality of the paper.
The two main problems: The fact that the contributions in comparison with GOHOME were not clear,
and the lack of results in comparable benchmark were both carefully addressed.

I am changing the score to a 6 provided that the authors proofread the paper as well recommended
by reviewer 4LEw, add the other benchmark results and better compare with GOHOME.

The paper does not qualify as of containing major new contributions. However, I think
the incremental contributions shown are sufficient for me to lean towards acceptance.

---

> ### Author Response · Authors · 2021-11-16
> **Response to Reviewer PgCY (Part 1)**
>
> Thanks a lot for your detailed and valuable feedback !
>
> >I could not find a clear distinction with literature
> While reading the related work section I found hard to contrast the ideas of the paper with the ones proposed on the literature. The difference between this method and GOHOME was not clear to me in this sense. From what I understood the main contribution here is the modality combination based on the collision free end-point sampling. However, this was not obvious for me when reading the paper.
>
> After revision, our main contributions are the design of a hierarchical heatmap generation tackling memory and computational cost problems of usual heatmap approaches, and the introduction of a recombination module that can generate scene-consistent prediction, taking as input the prediction of any marginal trajectory forecasting model.
>
> >The implemented baselines seemed to have performed way worse than expected.
> For example, both [1] and [7] are methods designed for joint prediction and scene level prediction. This is contradicted by the observations made by the authors on page 8, quote: " ... This problem is aggravated in the joint training case, since the modality selected is the same for all agents in a training sample....". Both [1] and [3] fairly confidently claim producing multimodal joint predictions. Either the version implemented by the authors does not correctly match the other works in the literature or there are specific findings on the limitations existent in other methods. Also, the paper lack comparison with latent variable based [2][5][[6] multi modal prediction methods. They seem to perform better when the issue is diverse prediction. Definitely the mode does not tend to be the same for all predictions in those cases.
>
> Actually, while capable of predicting for multiple agents at the same time, LaneGCN [1] never mentions multi-agent predictions nor evaluates on a multi-agent benchamark. We indeed notice that joint training degrades multimodality in our baselines, but remark that on the contrary SceneTransformer [2] doesn’t suffer the same problems thanks to its shared architecture between modalities.
>
> We aknowledge that our implementations of the baselines are not exactly similar to the original papers since neither released code, as we found more fair to have the same backbones between them (we also couldn't afford some of their training costs, as SceneTranformer takes 3 days of training on a TPU custom hardware accelerator, compared to 7 hours on a single 2080 RTX for us). We focused on implementing the key ideas of each approach to tackle multi-modality and scene-consistent prediction.
>
> Actually SceneTransformer [2] also notices in Appendix B1 and Table 10 a similar trade-off, where the joint model applied on the marginal benchmark performs much worse (37% MissRate instead of 22%). We actually implemented and compared with ILVM [3], which in their own paper already compared positively to MFP [4], and indeed since their training doesn’t have to specialize on either the marginal or the joint case they don’t suffer a trade-off between either but remain less accurate than other methods in our evaluation.
>
> >A single dataset evalution might not be enough for the community to access the value of this paper.
> Motion prediction is currently a very competitive scene. Even though I appreciate the idea on focusing reporting results on multi-agents and joint prediction tasks, it is important to also evaluate the prediction on more standard benchmarks such as argoverse. With results on argoverse, me and the rest of the community can more quickly access the the capabilities of the proposed heat map prediction pipeline, on producing consistent ego-agent motion prediction. I understand that this would not be the main results, but it would be helpful.
>
> We definetly agree, we focused on Interpret since it was the only benchmark focusing on multi-agent joint prediction, but following reviewers remarks we are currently evaluating on the standard Argoverse and NuScenes benchmarks, and trying to adapt NuScenes to the multi-agent case.
>
> Preliminary results on the validation sets of Argoverse and NuScenes give the following numbers (official submission on online leaderboard remains to be done):
>
> | Argoverse validation| minFDE6 |  MR6 |
> | -----------         | --------|----- |
> | GOHOME              | 1.26    |7.1   |
> | THOMAS              | 1.22    |6.6   |
>
> | NuScenes validation| minADE5  |  MR5 | minFDE1 |
> | -----------         | --------|----- |-----    |
> | GOHOME              | 1.42    | 57   |6.99     |
> | THOMAS              | 1.34    | 54   |6.73     |

---

> > ### Comment · Reviewer_PgCY · 2021-11-21
> > **Review Updated**
> >
> > Thanks for the carefully written response.
> > I have updated the review.

---

> ### Author Response · Authors · 2021-11-16
> **Response to Reviewer PgCY (Part 2)**
>
> >Computational cost is apparently not shown.
> I could not find the computational cost of running and training the proposed method on the paper. It seems that even with the optimizations made on the heat map estimation the training time and inference time should be high. Those kinds of motion prediction models have the necessity to further scale with way more data. Thus, highly complex methods can be hard to train. There is also the case of potential future execution in an embedded hardware
>
> We are also currently running evaluation on the computational cost in order to add the results in the paper.
>
> Preliminary results on training and inference time on a single 2080 RTX GPU give the following numbers  on the Interpret dataset with a batchsize of 16 (training lasts 16 epochs):
>
> | Computational cost| Inference time  |  Training time |
> | -----------         | --------|----- |
> | GOHOME              | 33 ms   | 12.8 hours   |
> | THOMAS              | 21 ms    | 7.5 hours  |
>
> (Note that these numbers are on the single agent track, and don't account for the decoding part having to be repeated multiple times in the multi-agent case. We will try to provide additional comparison for the multi-agent case.)
>
> We are working to get more experiments highlighting the trade-off and justifying our choices of parameters for the hierarchical heatmap output.
>
> >I am not sure about the scene modality vectors ( Section 3.2.2), are they like seed parameters which are going to be used to incorporate the joint consistent scene level ? I couldn't really find any sort of detailed explanation on that.
>
> The goal is to transition from agent modalities to scene-modalities. In order to do so, we create scene-modalitity vectors. From each scene-modaliy vector, we will choose a trajectory for each agent and a probability. The scene modality vectors are initialized from separate learned weights for each modality (they could also be initialized from one-hot encodings for simplicity's sake). They gain information from the context through cross-attention layers, and are then used to compute scalar product with each modality vector of one agent, in order to produce the linear combination modality for this agent. This is done for every agent. We are working on Fig. 4. and on Sec. 3.2.2 to make this part clearer.
>
> >Why the lack of ADE for scene level metrics or using similar metrics as [3] ?
>
> We didn’t report on ADE for brevity and since it is already pretty correlated with FDE, but we could add it in supplementary if you deem it necessary.
>
>
>
> [1] Liang, Ming, et al. "Learning lane graph representations for motion forecasting" In *ECCV*, 2020
>
> [2] Ngiam, Jiquan, et al. "Scene Transformer: A unified architecture for predicting multiple agent trajectories." arXiv preprint arXiv:2106.08417 (2021).
>
> [3] Casas, Sergio, et al. "Implicit latent variable model for scene-consistent motion forecasting." In *ECCV*, 2020.
>
> [4] Tang, Charlie, and Russ R. Salakhutdinov. "Multiple futures prediction." In *NeurIPS*, 2019

---

### Official Review · Reviewer_kDAs · 2021-11-02

**Correctness:** 3
**Technical Novelty And Significance:** 2
**Empirical Novelty And Significance:** Not applicable
**Recommendation:** 6
**Confidence:** 4

**Main Review:**

Strengths:
1. The paper is well written and easy to follow.
2. Good performance was achieved on the Interaction multi-agent prediction challenge.

Weaknesses:
1. The method largely follows previous works, with little technical contributions. And even for these limited technical novelties, many of them are not well supported with experimemts.
In the end of introduction section, the authors summarize three main contributions:

i) an efficient graph-based model. However, the implementation mainly follows LaneGCN and GOHOME.

ii) a collision-aware endpoint sampling. The algorithm mainly follows GOHOME, with a small modification in that when we sample an endpoint for actor a modality k, we not only mask out the same region for other modalities of actor a, but also mask out for other actors of modality k. However, this modification doesn't make sense to me, as we still have a scene-consistent modality recombination module afterwards, therefore the second mask-out won't make much difference here. I also don't see an ablation study for this specific change (ie, do not mask-out for other actors of modality k, which means exactly follows the original algorithm in GOHOME). I want to see this ablation for the last three rows in Table 3.

iii) a novel recombination model that could obtain scene-consistent trajectories across the agents. To me this is the core novelty of the paper. From the last two rows in Table 3 we can see that with the proposed recombination module joint metrics improve a lot. However, the marginal metrics degrade a lot (e.g., miss rate more than doubled). The only explanation is that with the proposed module, the same modality (also the sub-optimal one) of an actor is chosen for multiple times in scene-level multi-modalities. To me this may not be what we want: scene-level consistency, but sacrifices actor-level precision a lot. Further analysis is needed here to convince me that this trade-off is worthwhile.

2. Another technical novelty that's not mentioned is the hierarchical iterative refinement of endpoints from the heatmap. However, no ablation study is done on this. I'd like to see a speed-performance comparison with original endpoint estimation model (as in GOHOME).

3. Also, the overall runtime efficiency of the method is not given.

4. GOHOME is the closest approach but I don't see it as a baseline in the experiments.

**Summary Of The Paper:**

This paper addresses the task of multi-agent multi-modal trajectory prediction in the context of autonomous driving. Inspired by previous works, the problem is divided into first estimate the goal (end-points), and then re-construct the full trajectory. For end-point estimation, this paper proposes hierarchical iterative refinement from a probability heatmap, with collision-aware greedy sampling to generate collision-free multi-agent trajectories. To further produce scene-consistent multi-agent multi-modal trajectories, the paper also proposes a modality combination ranking module that re-orders the modality of each agent. The method is validated on the Interaction multi-agent prediction challenge and ranks 2nd on the multi-agent track (the 1st-rank entry was submitted 9 days after submission) and 1st on the conditional multi-agent track.

**Summary Of The Review:**

Despite good results on the leaderboard, given the limited technical novelty without solid supporting evidences, I do not recommend this paper for acceptance at current status.

**Updated review after rebuttal:**

Thanks to the authors for the revision that clarifies the technical novelity much better, and the additional experiments that compare with the GOHOME baseline thoroughly on more benchmarks. These address most of my concerns and therefore I raise my rating.

---

> ### Author Response · Authors · 2021-11-16
> **Response to Reviewer kDAs (Part 1)**
>
> Thanks a lot for your detailed and valuable feedback !
>
> > i) an efficient graph-based model. However, the implementation mainly follows LaneGCN and GOHOME.
>
> The efficiency and novelty mostly comes from producing the heatmap using a hierarchical process. We will better highlight this as a novelty and propose additional experiments to support it in the coming revision of the paper.
>
> > ii) a collision-aware endpoint sampling. The algorithm mainly follows GOHOME, with a small modification in that when we sample an endpoint for actor a modality k, we not only mask out the same region for other modalities of actor a, but also mask out for other actors of modality k. However, this modification doesn't make sense to me, as we still have a scene-consistent modality recombination module afterwards, therefore the second mask-out won't make much difference here. I also don't see an ablation study for this specific change (ie, do not mask-out for other actors of modality k, which means exactly follows the original algorithm in GOHOME). I want to see this ablation for the last three rows in Table 3.
>
> The ablation is actually available as the lines “Heat – Algo – Marg” (marginal version of the endpoint sampling algorithm which doesn't mask the other actors) and “Heat – Algo – Joint” (proposed version of collision-aware endpoint sampling), However, as the novelty of this proposed sampling algorithm is lesser, we will move it to a baseline as proposed by other reviewers for clarity.
>
> >iii) a novel recombination model that could obtain scene-consistent trajectories across the agents. To me this is the core novelty of the paper. From the last two rows in Table 3 we can see that with the proposed recombination module joint metrics improve a lot. However, the marginal metrics degrade a lot (e.g., miss rate more than doubled). The only explanation is that with the proposed module, the same modality (also the sub-optimal one) of an actor is chosen for multiple times in scene-level multi-modalities. To me this may not be what we want: scene-level consistency, but sacrifices actor-level precision a lot. Further analysis is needed here to convince me that this trade-off is worthwhile.\
>
> This is a good point. Actually SceneTransformer [1] notices a similar trade-off betweem their joint and marginal models (Appendix B1 and Tab. 10 in their paper). We therefore believe this is not a consequence of our method, but of the joint task itself.
>
> Short example: in a case with 2 agents, one agent A with two clear possible futures (A1-A2) evaluated at 50% probability each, and another agent B with one dominant future at 90% (B1) and another future at 10% (B2). Assuming independence between agents, if we can only propose 2 scene-modalities, the best combination is (A1-B1, A2-B1) with probability 0.5x0.9+0.5x0.9 = 0.9, compared to (A1-B1, A2-B2) with probability 0.5x0.9+0.5x0.1 = 0.5. We therefore have to sacrifice some modalities for agent B in order to get better scene-consistent modalities across all-agents.
>
> We believe the choices of trade-off between marginal and joint cases belong in a broader scope beyond this paper, as we mostly focus on proposing a solution for the joint case that was little explored so far. However, one highlight of our method is that it still generates marginal prediction in a first step, and only then derive joint prediction with very little additional cost, we therefore still could provide the marginal solution in the pipeline for later processing and decision-making.
>
> [1] Ngiam, Jiquan, et al. "Scene Transformer: A unified architecture for predicting multiple agent trajectories." arXiv preprint arXiv:2106.08417 (2021).
>
> >Another technical novelty that's not mentioned is the hierarchical iterative refinement of endpoints from the heatmap. However, no ablation study is done on this. I'd like to see a speed-performance comparison with original endpoint estimation model (as in GOHOME).
> Also, the overall runtime efficiency of the method is not given.
>
> We much agree with this assessment and are working to provide speed evaluations and trade-off studies for this part.
>
> Preliminary results on training and inference time on a single 2080 RTX GPU give the following numbers on the Interpret dataset with a batchsize of 16 (training lasts 16 epochs):
>
> | Computational cost| Inference time  |  Training time |
> | -----------         | --------|----- |
> | GOHOME              | 33 ms   | 12.8 hours   |
> | THOMAS              | 21 ms    | 7.5 hours  |
>
> We are working to get more experiments highlighting the trade-off and justifying our choices of parameters for the hierarchical heatmap output.

---

> ### Author Response · Authors · 2021-11-16
> **Response to Reviewer kDAs (Part 2)**
>
> >GOHOME is the closest approach but I don't see it as a baseline in the experiments.
>
> As queried by other reviewers, we are working to evaluated THOMAS on more broadly used datasets such as Argoverse and NuScenes, and will provide comparisons with GOHOME on these.
>
> Preliminary results on the validation sets of Argoverse and NuScenes give the following numbers (official submission on online leaderboard remains to be done):
>
> | Argoverse validation| minFDE6 |  MR6 |
> | -----------         | --------|----- |
> | GOHOME              | 1.26    |7.1   |
> | THOMAS              | 1.22    |6.6   |
>
> | NuScenes validation| minADE5  |  MR5 | minFDE1 |
> | -----------         | --------|----- |-----    |
> | GOHOME              | 1.42    | 57   |6.99     |
> | THOMAS              | 1.34    | 54   |6.73     |

---

### Official Review · Reviewer_4LEw · 2021-11-03

**Correctness:** 3
**Technical Novelty And Significance:** 2
**Empirical Novelty And Significance:** 2
**Recommendation:** 6
**Confidence:** 4

**Main Review:**

- The overall idea of iterative hierarchical heatmap refinement is interesting and appears to be novel. The experiments demonstrate the ideas validity, although there is definitely room for further strengthening.

- The related work is thorough, one quick comment here is to also reference the following work in the discussion of methods which can predict multiple agents at the same time (beginning of the last paragraph before Section 3): "MATS: An Interpretable Trajectory Forecasting Representation for Planning and Control" by B. Ivanovic, A. Elhafsi, G. Rosman, A. Gaidon, M. Pavone in CoRL 2020.

- Acronyms should be explained when they are first used. For instance, it might not be obvious what a UGRU is in Section 3.1.1. What does MR optimization mean in Section 3.1.3?

- Do not use contractions in scientific writing (e.g., change "don't" to "do not").

- A thorough round of editing is necessary, there seem to be writing errors that materially change the meaning of sentences. For instance, "The final output is therefore no strictly a re-ordering..." Should this have said "not strictly"? Same question for "which is JointMR where colliding modalities are also counted as misses even if they are closer than the defined threshold." Should this have just said "where colliding modalities are also counted as misses?" It is not very clear what these metrics mean here.

- The novelty with respect to the original GOHOME work needs to be made clearer, there seem to be many reused components and it is difficult to tell which parts are new in this work.

- Related to the above point, it is a bit difficult to understand this work's core contributions in Sections 3.2.1 and 3.2.2, also because the writing makes this difficult (e.g., a spurious use of "However" makes it difficult to determine if a statement is being negated or not).

- There are some key insights missing from the experimental section, which will be expanded on below.

- Ultimately, the method is only evaluated on one dataset, and it is difficult to determine if its performance would generalize to other datasets (for instance, what about nuScenes, Lyft, Waymo Open, Argoverse, etc?).


**Summary Of The Paper:**

This work proposes THOMAS, a hierarchical heatmap refinement scheme for multi-agent trajectory forecasting. At its core lies a GOHOME-style architecture with a few modifications (predicting only endpoints, iteraetively improving the heatmap output, collision-free output sampling, modality combination re-ranking) that yield scene-consistent predictions, which are demonstrated to yield better performance across a wide-variety of metrics on the Interaction dataset.

**Summary Of The Review:**

While the overall idea of iterative hierarchical heatmap refinement is interesting, there are some significant areas of improvement that make it difficult to argue for the acceptance of this work as-is. For instance:
- Writing errors in key areas make it difficult to parse what is going on internally in the model.
- The novelty with respect to the original GOHOME model should be expanded upon, perhaps with a concrete sentence stating the core differences near the beginning of Section 3 and stating the relationship of this work to GOHOME in the related work.
- The experiments are good at showing that the model works, but there are a bit of key insights missing. For example, why are only two rounds of heatmap upsampling done? What do the performance curves look like with increased points? What does the Pareto front look like with respect to computation time or required FLOPs (at least to indicate to a reader why stopping at two levels makes sense)?
- Finally, another dataset would really strengthen this work and aid in convincing readers that the method's performance extends beyond the Interaction dataset.

Addressing these would certainly ready this paper for publication in this or a similar venue.

### Post-Rebuttal

The added results and comments in the author rebuttal address most of my concerns. I share the same overall sentiment as Reviewer PgCY, the paper may not have a single "major" novelty, but the set of presented incremental contributions are sufficient to convince me of the performance and utility of this work over prior approaches, and I believe this paper is now more suitable for publication.

---

> ### Author Response · Authors · 2021-11-16
> **Response to Reviewer 4LEw**
>
> Thanks a lot for your detailed and valuable feedback !
>
> >Acronyms should be explained when they are first used. For instance, it might not be obvious what a UGRU is in Section 3.1.1. What does MR optimization mean in Section 3.1.3?
>
> U-GRUS are a simple asymmetrical variant of bidirectional GRUs, where a backward RNN pass is done before the forward pass and instead of in parallel. We added the reference paper [1] in the bibliography. MR optimization means MissRate optimization and refers to the endpoint sampling algorithm from HOME [2], detailed with modifications in 3.2.1, we will rewrite this in the paper for better clarification.
>
> >A thorough round of editing is necessary, there seem to be writing errors that materially change the meaning of sentences. For instance, "The final output is therefore no strictly a re-ordering..." Should this have said "not strictly"? Same question for "which is JointMR where colliding modalities are also counted as misses even if they are closer than the defined threshold." Should this have just said "where colliding modalities are also counted as misses?" It is not very clear what these metrics mean here.
>
> Indeed your interpretations are correct, we are revising this in the paper and proofreading it more thoroughly.
>
> >The novelty with respect to the original GOHOME work needs to be made clearer, there seem to be many reused components and it is difficult to tell which parts are new in this work.
>
> In short, the encoding part is the same as GOHOME. The heatmap generation differs by using attention to a grid instead of lane rasters, and we also propose a hierarchical process for more efficient decoding. The endpoint sampling is also similar, with the exception of the collision-free variant detailed in 3.2.1 which we will move as a baseline in the revised version for clarity.
>
> >The experiments are good at showing that the model works, but there are a bit of key insights missing. For example, why are only two rounds of heatmap upsampling done? What do the performance curves look like with increased points? What does the Pareto front look like with respect to computation time or required FLOPs (at least to indicate to a reader why stopping at two levels makes sense)?
>
> We are running more extensive experiments to produce computation time numbers and better analysis of the performance – computational cost trade-off.
>
> Preliminary results on training and inference time on a single 2080 RTX GPU give the following numbers on the Interpret dataset with a batchsize of 16 (training lasts 16 epochs):
>
> | Computational cost| Inference time  |  Training time |
> | -----------         | --------|----- |
> | GOHOME              | 33 ms   | 12.8 hours   |
> | THOMAS              | 21 ms    | 7.5 hours  |
>
> (Note that these numbers are on the single agent track, and don't account for the decoding part having to be repeated multiple times in the multi-agent case. We will try to provide additional comparison for the multi-agent case.)
>
> We are working to get more experiments highlighting the trade-off and justifying our choices of parameters for the hierarchical heatmap output.
>
>
>
> >Finally, another dataset would really strengthen this work and aid in convincing readers that the method's performance extends beyond the Interaction dataset.
>
> Following reviews we are also running additional trainings and evaluations on Argoverse and NuSCenes to provide addition comparison points on more broadly used datasets.
>
> Preliminary results on the validation sets of Argoverse and NuScenes give the following numbers (official submission on online leaderboard remains to be done):
>
> | Argoverse validation| minFDE6 |  MR6 |
> | -----------         | --------|----- |
> | GOHOME              | 1.26    |7.1   |
> | THOMAS              | 1.22    |6.6   |
>
> | NuScenes validation| minADE5  |  MR5 | minFDE1 |
> | -----------         | --------|----- |-----    |
> | GOHOME              | 1.42    | 57   |6.99     |
> | THOMAS              | 1.34    | 54   |6.73     |
>
> [1] Rozenberg, Raphaël, Joseph Gesnouin, and Fabien Moutarde. "Asymmetrical Bi-RNN for pedestrian trajectory encoding." arXiv preprint arXiv:2106.04419 (2021).
>
> [2] Gilles, Thomas, et al. "HOME: Heatmap output for future motion estimation" In *ITSC*, 2021.

---

> > ### Comment · Reviewer_4LEw · 2021-11-28
> > **Updated Review**
> >
> > Thank you for the detailed response! I apologize for not posting a reply earlier.
> >
> > These added results and comments address most of my concerns. I share the same overall sentiment as Reviewer PgCY, the paper may not have a single "major" novelty, but the set of presented incremental contributions are sufficient to convince me of the performance and utility of this work over prior approaches, and I believe this paper is now more suitable for publication. Thank you again for putting in the effort to address our comments!

---

### Author Response · Authors · 2021-11-21
**Paper revision**



We thank a lot again the reviewers for their insightfull feedback. We updated a revised paper where we tried to include the comments and suggestions.

Main changes are the following:

- We revised the main contributions to the hierarchical heatmap generation and the recombination module, moving the collision-free sampling to a baseline in the ablation study
- We elaborated on the main common parts and differences with GOHOME in the Related work and the beginning of Section 3.
- We also added the suggested references in the Related work
- We rewrote the description the hierarchical heatmap decoding method in Sec. 3.1.2 in a more general fashion.
- We updated Figure 4 (recombination module) and tried to make it more clear and helpful for the corresponding text description. We also rewrote the recombination explanation in Sec. 3.2 to make it more clear and describe better the intuition behind the method.
- We added experiments on the hierarchical heatmap generation in Table 3 to demonstrate the speed gains compared to GOHOME. Here is a summary of the times for training and simultaneous multi-agent prediction:

| Computational cost|   Training time | Inference 32 agents  |Inference 128 agents  |
| -----------         |----- |----- |----- |
| GOHOME              |  12.8 hours   |36 ms | 90 ms|
| THOMAS              | 7.5 hours  | 20 ms | 31 ms|
- We added evaluation on the Argoverse and NuScenes single agent benchmarks in the Appendix to give a better appraisal of the performance of our model. Here is a summary of our numbers compared to SOTA on the online test leaderboards (available at https://eval.ai/web/challenges/challenge-page/454/leaderboard/1279# and https://eval.ai/web/challenges/challenge-page/591/leaderboard/1659):

| Argoverse test| minFDE6 |  MR6 |
| -----------         | --------|----- |
| LaneGCN             | 1.36    |16.3   |
| AutoBot             | 1.41    |16   |
| TNT             | 1.54    |13.3   |
| SceneTranformer     | **1.23**    |12.6   |
| LaneRCNN             | 1.45    |12.3   |
| DenseTNT            | 1.49    |10.5   |
| GOHOME              | 1.45    |10.5   |
| HOME                | 1.44    |**9.8**   |
| **THOMAS**              | 1.44    |10.4   |

| NuScenes leaderboard| minADE5  |  MR5 | minFDE1 |
| -----------         | --------|----- |-----    |
| AgentFormer        | 1.86    | _   |_     |
| WIMP              | 1.84    | 55   |18.49     |
| P2T              | 1.45    | 64   |10.50     |
| GOHOME              | 1.42    | 57   |6.99     |
| AutoBot              | 1.37    | 62   |8.19     |
| PGP                 | **1.30**    | 57   | 7.72   |
| **THOMAS**              | 1.33    | **55**   |**6.71**     |

---

> ### Author Response · Authors · 2021-11-22
> **Additional experiment**
>
> We have just uploaded a last revision, where we included an additional experiment on hierarchical refinement in the Appendix. As Reviewer 4LEw wanted to have more information about the performance curve, and Reviewer ZJY1 was worried that only using the last iteration of refinement might miss some coverage-critical points, we displayed the performance trade-off between inference time and MissRate$_6$ in Figure 7. We  set the number N of upsampled points at the last refinement iteration from 2 to 128 and plotted the resulting curve. We cannot display the figure here but results can be summarized by the following table:
>
> | N| 2 | 4 | 8 | 16 | 32 | 64 | 128 |
> | ------| ------ | ------ | ------ | ------ | ------ | ------ | ------ |
> |Inference time (ms) |   39.8| 40.1 | 40.6 | 41 | 43 | 46 | 52 |
> |MissRate$_6$ |  10.1| 6.4 | 4.39 | 3.78 | 3.76| 3.76 | 3.76 |
>
> From N=16 and lower, coverage performance starts to degrade while little speed gains are made. For our final model, we chose a relatively high N=64 as the speed loss was still reasonable and we wanted to insure a wide coverage.

---

### Decision · Program_Chairs · 2022-01-20

**Decision:**

Accept (Poster)

**Comment:**

This paper proposes a joint multi-agent trajectory prediction framework for multiple agents using a "heatmap" estimation approach employing a hierarchical strategy and sparse image generation for for efficient inference. The method takes a set of predicted trajectories for each agent produces reorderings. The work yields a top result on a competitive leaderboard.

While multiple reviewers were initially concerned about the paper not making a single major contribution, the author response discussion helped to clear up the degree of novelty. Further experiments provided during the review also led to multiple reviewers increasing their score. In the end, all reviewers recommend acceptance of this paper.

As such the AC recommends accepting this paper.